# Cadmium Elimination via Magnetic Biochar Derived from Cow Manure: Parameter Optimization and Mechanism Insights

Yi Wen [1,2,†], Dingxiang Chen [1,2,†], Yong Zhang [1,2], Huabin Wang [1,2,*] and Rui Xu [1,2,*]

1 School of Energy and Environment Science, Yunnan Normal University, Kunming 650500, China; wyaquarius@foxmail.com (Y.W.); ynnuchendx@foxmail.com (D.C.); yongzhang7805@126.com (Y.Z.)
2 Yunnan Key Laboratory of Rural Energy Engineering, Kunming 650500, China
* Correspondence: hbwang@ynnu.edu.cn (H.W.); ecowatch_xr@163.com (R.X.); Tel./Fax: +86-27-65940928 (R.X.)
† These authors contributed equally to this work.

**Abstract:** Designing an efficient and recyclable adsorbent for cadmium pollution control is an urgent necessity. In this paper, cow manure, an abundant agricultural/animal husbandry byproduct, was employed as the raw material for the synthesis of magnetic cow manure biochar. The optimal preparation conditions were found using the response surface methodology model: 160 °C for the hydrothermal temperature, 600 °C for the pyrolysis temperature, and Fe-loading with 10 wt%. The optimal reaction conditions were also identified via the response surface methodology model: a dosage of 1 g·L$^{-1}$, a pH of 7, and an initial concentration of 100 mg·L$^{-1}$. The pseudo-second-order model and the Langmuir model were used to fit the Cd(II) adsorption, and the adsorption capacity was 612.43 mg·g$^{-1}$. The adsorption was dominated by chemisorption with the mechanisms of ion-exchange, electrostatic attraction, pore-filling, co-precipitation, and the formation of complexations. Compared to the response surface methodology model, the back-propagation artificial neural network model fit the Cd(II) adsorption better as the error values were less. All these results demonstrate the potential application of CM for Cd(II) removal and its optimization through machine-learning processes.

**Keywords:** magnetic biochar; cow manure; cadmium removal; response surface methodology; artificial neural network

## 1. Introduction

The Cd-containing effluent is a severe issue that endangers human health and the environment. Cd(II) can accumulate in cereal grains when wastewater flows into farmland [1], and an intake of 0.125 mg of Cd per kg$^{-1}$ of body weight can lead to severe health problems [2], such as renal illness, cancer, and bone damage. Researchers use biological, chemical, and physical adsorption as well as other methods to control Cd(II) contamination. It has been reported that the exogenous application of salicylic acid decreased the adverse effects of Cd(II) on photosynthesis in maize [3]. Studies also showed that an increased exogenous supply of metal ions, e.g., Fe(II), lead to decreased Cd(II) uptake in rice and Arabidopsis [4,5].

As adsorbents, biomaterials have several advantages, including being low-cost and environmentally friendly and having efficient phosphorous-removal abilities [6]. Researchers have prepared biochar derived from different biomass materials for Cd(II) adsorption, such as corn cob, rice husk [7], and tobacco stalk [8]. However, absorption efficiency needs to be improved, as the biochar can only be separated from wastewater and recycled with difficulty [9,10]. As a matter of fact, biochar only achieves adsorption rather than removal. In order to tackle this problem, diverse types of magnetic biochar, which has the capability of adsorbing heavy metals efficiently and can be separated from water using an external magnetic field, were designed. The typical techniques for synthesizing magnetic biochar

include liquid reductio, impregnation, and precipitation methods; in addition, one of the most common modification techniques is to load iron onto biochar [6].

Manure production increased dramatically as a result of the recent rapid expansion of concentrated animal feeding operations in many parts of the world [11]. It is imperative to find solutions for the processing and usage of cow manure (CM) produced in these sizable feedlots [12]. The CM biochar formed after pyrolysis has a plentiful pore structure; thus, CM biochar has the potential for heavy metal adsorption [13].

The adsorption effect of CM biochar on heavy metals can be enhanced via modifications, such as via acid or metal oxide. Additionally, during the preparation of magnetic CM biochar, three factors, namely pyrolysis temperature (PT), hydrothermal temperature (HT), and iron content, are key [14]. Previous research has demonstrated that the specific surface area (SSA) and the functional groups on the surfaces of biochar are disparate when preparation conditions are different. Meanwhile, the increase in Cd(II) adsorption capacity can be attributed to the increased amount of SSA and functional groups, especially phenolic hydroxyl [15,16]. Some researchers have explored the influence of a single factor on adsorption capacity, but there are few researchers focusing on mutual interference between the three factors mentioned above. Moreover, the optimum preparation conditions of magnetic CM biochar used for Cd(II) adsorption have not been summarized. Similarly, during adsorption, three factors, namely dosage, pH, and initial concentration, are key [17], and their combined effects on Cd(II) adsorption has also not yet been clarified [18,19]. Therefore, it is necessary to discuss the relationship between multiple parameters through model analysis in order to optimize the preparation and adsorption.

In traditional methods of optimization, only one parameter is varied at a time, which makes experimentation time-consuming and costly. Model fitting based on certain data can simplify the experimental design, thus reducing costs and saving time. The response surface methodology (RSM) adopts a reasonable experimental design method and obtains data through the test. By constructing a multiple quadratic regression equation to fit the function relationship between factors and response value, the optimal process parameters were sought by analyzing the regression equation [20]. The back-propagation artificial neural network (BP–ANN) is a soft computing technique that studies the process through the modification of network weights to produce the required response [21]. BP–ANN is better than the traditional nonlinear model when dealing with the complex relationship between the input and output variables [22].

However, the RSM and BP–ANN models have some disadvantages. When using the RSM model, the preparation and reaction conditions must be within the range of the test values. Moreover, there are nonlinear and uncertain relationships among the factors affecting the adsorption effect, so it can be difficult to predict the strengths of specific multivariate nonlinear function forms. Analogously, the BP–ANN model requires a large number of experiments for training. Currently, there is no method for determining the minimal number of experiments required for BP–ANN training; therefore, this methodology is troublesome for the design of experiments [23]. Although the two models have been widely used in adsorption studies, the prediction results are different. Some researchers found that the RSM model fitted better [24], while others found that BP–ANN fitted better in terms of the adsorption of heavy metals [25].

This study used CM as a raw material and utilized a combined hydrothermal and pyrolysis preparation [26], along with an added iron source ($FeSO_4$ and $FeCl_3$) for magnetic modification, to prepare the magnetic CM biochar, which was inexpensive and recyclable. In this study, RSM and BP–ANN models were utilized for the analysis of the adsorption effects of various types of magnetic CM biochar, and the preparation method of the biochar with the best adsorption effect was thus obtained. RSM and BP–ANN were also used to evaluate the adsorption effects under different reaction conditions, and the optimum reaction conditions of Cd(II) adsorption were thus identified. The adsorption kinetics, thermodynamics, and mechanism of Cd(II) were also investigated. The work can improve the adsorption efficiency, and provide ideas for the high-value utilization of CM, which

provides a reference for the adsorption of Cd(II). Meanwhile, the prediction effects of RSM and BP–ANN on the adsorption of Cd(II) by magnetic biochar were compared.

## 2. Experiments and Models

### 2.1. Chemicals and Materials

The primary chemicals, e.g., $FeCl_3$, $FeSO_4 \cdot 7H_2O$, NaOH, and $Cd(NO_3)_2$, used in this study were purchased from Sinopharm Co., Ltd. (Shanghai, China) and used without further purification. CM was collected from the Laohadu Demonstration Ranch in Mengzi City, Yunnan Province, China. The CM was oven–dried at 85 °C for 24 h and sieved through 100–mesh.

### 2.2. Synthesis of Magnetic CM Biochar

As shown in Figure 1, the magnetic CM biochar was synthesized in three parts. Chemical synthesis modification was the first step. The original CM (4, 6, and 8 g), 1.12 g $FeSO_4 \cdot 7H_2O$, 2.19 g $FeCl_3 \cdot 6H_2O$, and 38 mL NaOH (1 mol·L$^{-1}$) were mixed in the 100 mL deionized water. The three mixtures were severally sonicated for 30 min and stirred for 2 h. Hydrothermal carbonization was the second step. The three mixtures were placed in hydrothermal synthesis reactors separately and heated at different HT for 24 h (HT = 120, 160, and 200 °C). The carbonization was the final step. Three hydrothermal products were carbonized under nitrogen flow in a tube furnace at setting PT for 1 h to obtain three kinds of magnetic modified CM biochar (PT = 400, 600, 800 °C). The $Fe_3O_4$ or $Fe_2O_3$ nanoparticles had loaded on the products, and the products were marked as MCB$a$–$b$–$c$, where $a$ represented HT (120, 160, and 200 °C), $b$ represented PT (400, 600, and 800 °C), and $c$ represented iron content (5%, 10%, and 15%), and the magnetic CM biochar that possessed the most ideal characteristics was marked as MCB.

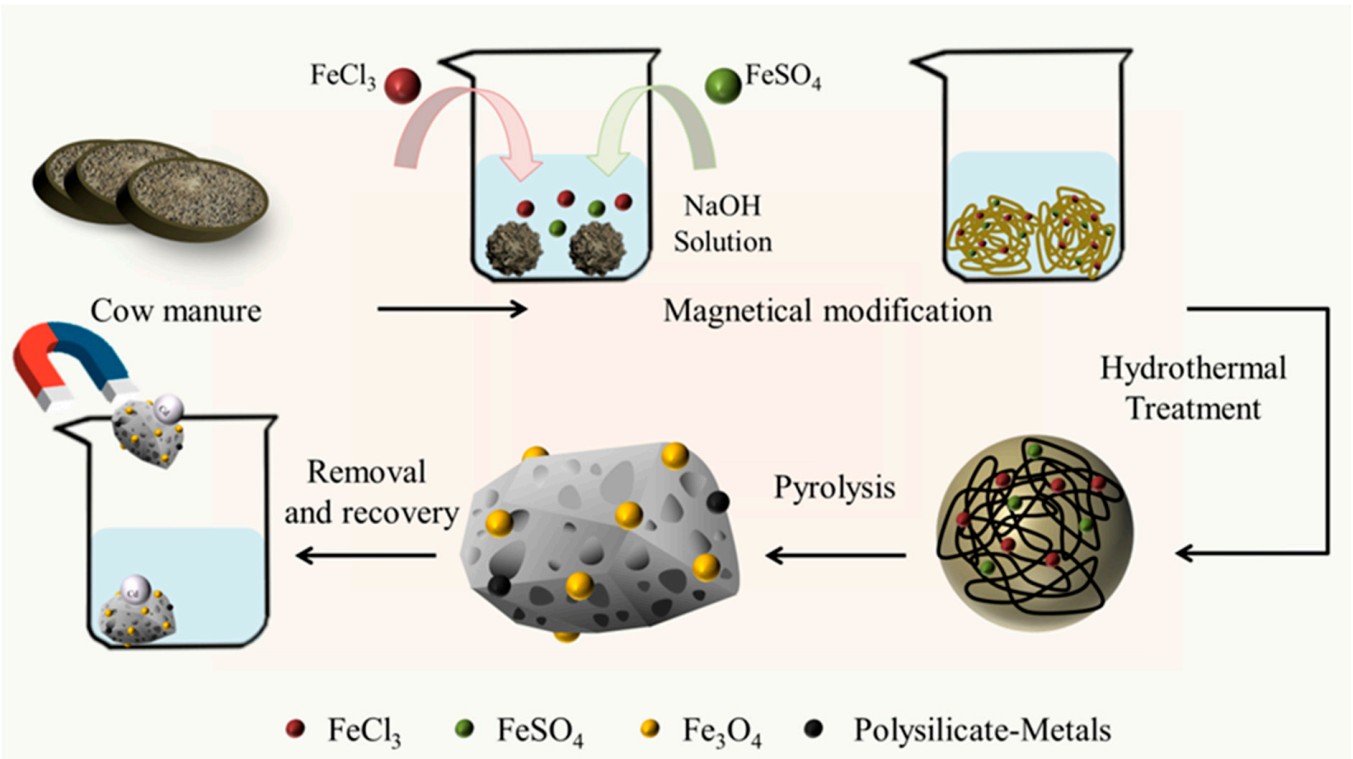

**Figure 1.** Preparation process for magnetic CM biochar.

### 2.3. Application of RSM

Box–Behnken design model was used to optimize synthesis parameters with Cd(II) adsorption capacity of magnetic CM biochar as the response and examine the relationship between various synthesis factors or reaction conditions. In the first part, RSM studied the effects of PT ($X_1$), HT ($X_2$), and iron content ($X_3$) on the adsorption capacity (Y) when the initial Cd(II) concentration was 100 mg·$L^{-1}$. In the second part, RSM studied the effects of dosage ($X_1$), pH ($X_2$), and the initial concentration ($X_3$) on the removal rate of Cd(II) (Y). The experimental findings were examined and fitted to a quadratic equation using Design Expert.8.0.

### 2.4. Application of BP–ANN

BP–ANN is an information processing system designed to mimic the analytic and processing functions of the human brain, and it uses supervised learning technology and trains by minimizing the square error of the network's output [21]. The BP–ANN model typically consists of an input layer, one or more hidden layers, and an output layer. In this work, one of the three-layer BP–ANN models were established for optimum preparation parameters, and the other for optimal adsorption conditions, respectively, and they were both established by MATLAB R2018b.

To optimize the preparation conditions of MCB, a BP–ANN model was established, whose quantities of neurons in the input layer and output layer were three and one, respectively. The input variables were PT, HT, and iron content, while the output variable was adsorption capacity. In order to explore the optimal reaction conditions of Cd(II) adsorption by MCB, another BP–ANN model was established. The number of neurons in the input layer was five, and the input parameters were dosage, pH, initial concentration, temperature, and time; the output parameter was the removal rate of Cd(II). The selection formula of the optimal number of neuron nodes in both hidden layers is shown in Equations (1)–(3) [27].

$$I < n - 1 \tag{1}$$

$$I < \sqrt{m + n} + \alpha \tag{2}$$

$$I = \log_2 n \tag{3}$$

where $n$ represents the number of neuronal nodes in the input layer, $I$ represents the number of neuronal nodes in the hidden layer, $m$ represents the number of neuronal nodes in the output layer, and $\alpha$ represents a constant between 0 and 10.

In this work, the data were selected as a training data group (Training), a verification data group (Verification), and a test data group (Test) according to the proportion of 75%, 15%, and 15%, respectively. The BP–ANN models were accepted when their correlation coefficients were higher than 0.95.

### 2.5. Adsorption Experiments

To obtain the MCB, 1 g·$L^{-1}$ of the various magnetic CM biochar was added into Cd(II) solution, respectively (20 mL, 100 mg·$L^{-1}$). And in the following adsorption experiment, MCB was used as an adsorbent to explore the optimal reaction conditions and adsorption mechanism of Cd(II). To obtain the optimal reaction conditions, different dosages of MCB (0.5, 1, and 1.5 g·$L^{-1}$) were placed in Cd(II) solutions with different initial concentrations (50, 100, and 150 mg·$L^{-1}$) and pH (5, 7, and 9), respectively.

For the adsorption kinetics, the MCB (1 g·$L^{-1}$) was added into the Cd(II) solution (20 mL, 100 mg·$L^{-1}$). After the set adsorption time, the supernatants were collected to quantify the residual content of Cd(II). For isotherm studies, the MCB (1 g·$L^{-1}$) was added to a series of Cd(II) solutions (100–1000 mg·$L^{-1}$). For the effect of pH on adsorption, the MCB (1 g·$L^{-1}$) was added into different solutions (pH = 1, 3, 5, 7, 9, and 11). The calculation

of adsorption capacity was presented in Text S1. All experiments were repeated three times, and the data were averaged with the actual data of the three times.

### 2.6. Analysis Methods

The elemental composition of biochar was analyzed with an elemental analyzer (EA) (Elementar Vario EL cube, Langenselbold, Germany). The morphology was analyzed with scanning electron microscopy (SEM) at 5 kV (Mira LMS, Tescan, Brno, Czech Republic). The SSA of biochar was detected by N2 adsorption isotherms at 77 K using a Micropore Analyzer (ASAP 2460, Micrometrics, Norcross, GA, USA). The crystal structure was characterized by X-ray diffractometry (XRD) (D8 Advance Sox−l, Bruker Co., Ltd, Billerica, MA, USA), and all samples were scanned over the region of 10–80°. The electron binding energy and elemental valence were analyzed using X-ray photoelectron spectroscopy (XPS) (K–Alpha, Thermo Scientific, Waltham, MA, USA). The functional groups were qualitatively examined using a Fourier transform infrared spectrometer (FTIR) (Niolet iN10EA, Thermo Scientific). The contents of the elements in the solutions were determined by inductively coupled plasma mass spectrometry (ICP) (Thermo Fisher–X series, Waltham, MA, USA).

## 3. Results and Discussion

### 3.1. Preparation Conditions Optimization

### 3.1.1. RSM Analysis

Experimental factors were summarized in Table S1, and the actual values of adsorption quantity were summarized in Table S2. According to the results obtained by the regression analysis (Table S3), the R squared ($R^2$) was calculated as 0.9620, indicating the model was significant. The *p*-value and *F*-value were 0.0004 (significant) and 1.75 (not significant), respectively, indicating the model was valid [20]. Shown in Equation (4), there was a quadratic polynomial equation of the relationship between the response value and the three experimental parameters:

$$Y = 73.7295 + 0.02475 \, X_1 + 0.22355 \, X_2 + 28.9042 \, X_3 + 8.9262 \, X_1 \, X_2 + 0.2112 \, X_2 \, X_3 - 8.07 \, X_1 \, X_3 - 2.93 \, X_1^2 - 1.02 \, X_2^2 - 261.45 \, X_3^2 \tag{4}$$

As the mutual effects between variables according to Equation (4), the relationships between the response value and the three parameters can be represented by the contour maps. The two-dimensional contour maps and three-dimensional response surfaces obtained by RSM were presented in Figure 2. The adsorption capacity was sensitive to the three parameters. Within a certain range of PT (<600 °C), the increase in adsorption capacity of the various magnetic CM biochar could be seen with the increasing PT. This indicated that the increasing PT promoted the SSA of biochar, which provided potential adsorption sites for Cd(II) [28]. Accordingly, the increasing PT may improve the physical adsorption effect of magnetic CM biochar via pore–filling. Nevertheless, the adsorption capacity was weakened when the PT was higher than 600 °C, which may be because the complexation ability was weakened. The high PT can effectively attenuate the existence of oxygen-contained functional groups [29]. Meanwhile, as shown in Figure 2f, the contour plot tended to a circle, indicating the interaction between PT and iron content might be negligible [30]. According to the analysis of two-dimensional contour maps and three-dimensional response surfaces, the optimal strategy for preparing MCB, the adsorbent for Cd(II), can be derived. Considering the adsorption capacity for Cd(II) and preparation efficiency, the optimum parameters were procured as follows: PT = 600 °C, HT = 160 °C, and iron content = 10 wt%.

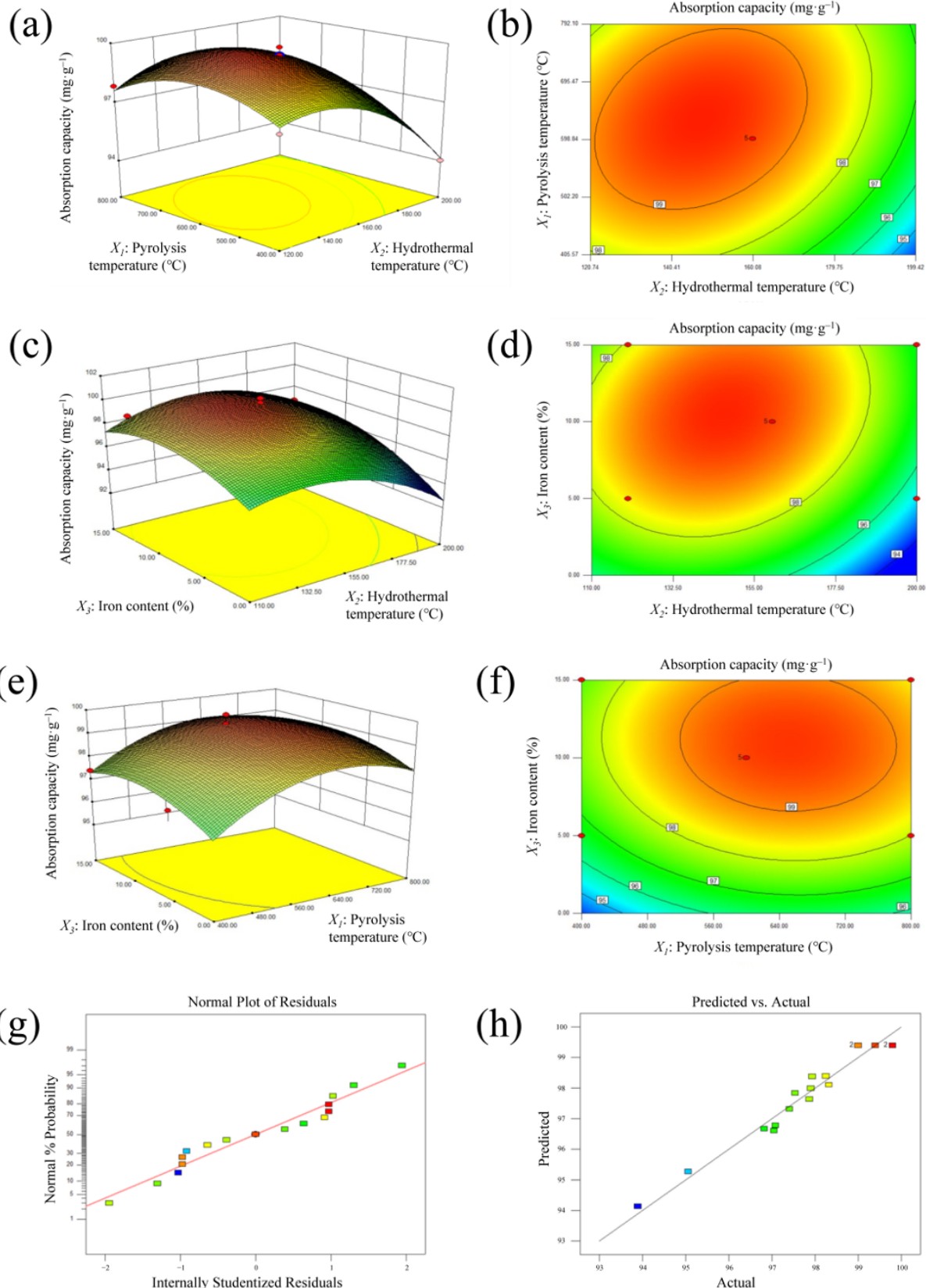

**Figure 2.** Duel effect on Cd(II) removal rate, Normal Plot of Residuals, and Predicted vs. Actual values: response surfaces (**a,c,e**), contour plots (**b,d,f**), Normal Plot of Residuals (**g**), and Predicted vs. Actual values (**h**). Experimental conditions: [initial concentration of Cd(II)] = 100 mg·L$^{-1}$, [dosage] = 1 g·L$^{-1}$, [time] = 24 h, [temperature] = 25 °C.

### 3.1.2. BP–ANN Model

The Levenberg–Marquardt training algorithm was used in the training of the data set, and the Training of the BP–ANN model was done until the error between the experimental and predicted values of responses reached the minimum. The weights and biases were all together known as neural network parameters. The trained model was validated via Validation (experimental data that were not used for training). The development of the BP–ANN model was carried out by dividing the data set into three groups: 70% for Training, 15% for Testing, and 15% for Validation. The weight values of the synaptic joints between the input and hidden layers, and that between the hidden and output layers, were calculated by well versed BP–ANN model for optimization.

The BP–ANN model used in this study (Figure 3a) had three input parameters (PT, HT, and iron content) and one output parameter, and the number of neurons varied from two to ten for the hidden layer. The Training, Validation, and Test showed that the BP–ANN model fitted well ($R^2 > 0.9$); thus, the accepted function had a great correlation coefficient between the target and simulated output values in Training, Validation, and Test (Figure 3d–g). The BP–ANN model tended to be stable at epoch 4, and the model had both a quick contingency speed and good stability.

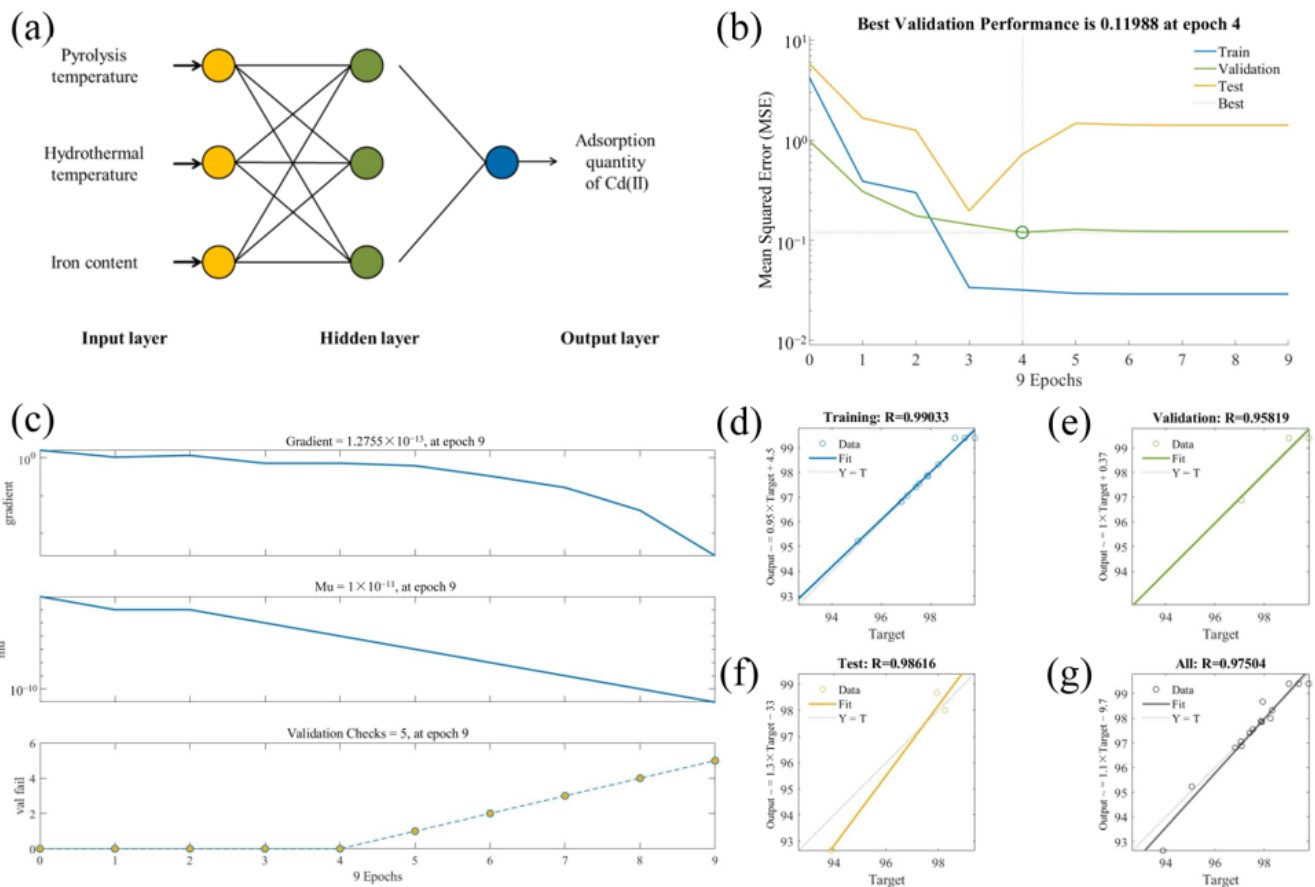

**Figure 3.** BP–ANN model of experimental data. Architecture of the BP–ANN model (**a**), error convergence curve during the iteration training of BP–ANN model (**b**), comparison between the output of BP–ANN model and the measured value (**c**), correlation between the experimental and simulated rate of removal during BP–ANN Training, Test, and Validation by magnetic CM biochar (**d**–**g**).

### 3.2. Reaction Conditions Analysis

#### 3.2.1. RSM

Experimental factors were summarized in Table S1, and the actual values of adsorption quantity were summarized in Table S4. Similarly to the previous analysis, the $R^2$ was 0.9924, and the *p*-value and Lack of Fit of the model were <0.0001 and 102.1500, respectively, indicating that the model was valid. In this RSM model, the values of Adeq Precision and C.V.% were 37.83 and 0.25%, respectively. Thus, the RSM model could also fit the data well (Table S5). The mathematical relationships between the response value and the three experimental parameters can be quantified by the following formula, Equation (5).

$$Y = 95.64 + 1.5\,X_1 + 1.99\,X_2 - 0.51\,X_3 \tag{5}$$

The two-dimensional contour maps and three-dimensional response surfaces obtained by RSM model were presented in Figure 4. The removal rate of MCB was strongly sensitive to pH and dosage, while the initial concentration had little effect. According to RSM model, the optimal reaction conditions could be worked out. Considering the removal rate of Cd(II), the optimal reaction conditions were procured as follows: Dosage = 1 g·L$^{-1}$, pH = 7, and initial concentration = 100 mg·L$^{-1}$.

#### 3.2.2. BP–ANN Model

The BP–ANN model used in this study (Figure 5a) had five input parameters (dosage, pH, initial concentration, temperature, and time), one hidden layer with five neurons, and one output parameter. The BP–ANN model fitted well ($R^2 > 0.9$); thus, the accepted function had a great correlation coefficient between the target and simulated output values in Training, Validation, and Test (Figure 5d–g). The BP–ANN model tended to be stable at epoch 21; hence, the BP–ANN model used in this work had both a quick contingency speed and good stability [31].

### 3.3. Comparison of the Developed RSM and BP–ANN Models

The performances of the RSM and BP–ANN models were compared using statistical parameters, such as $R^2$ and root mean square error (RMSE). For exploring the optimum preparation conditions of MCB, it can be seen from Table S6 that the BP–ANN model had a higher $R^2$ value and lower RMSE value compared to those of the RSM model (0.98 vs. 0.96 and 1.13 vs. 4.17). This indicated that the BP–ANN model had better-predicting ability than the RSM model for the investigation of optimum preparation conditions. For exploring the reaction conditions, both models had a good fitting effect for their $R^2$ values, both 0.99, while the BP–ANN model had a lower RMSE value compared to the RSM model (1.02 vs. 5.90).

As shown in Figure 6, the experimental values were plotted against the predicted values of the RSM model and BP–ANN model. It can be seen that the BP–ANN model performed better than the RSM model in exploring the optimum preparation conditions and reaction conditions. In terms of data fitting, the BP–ANN model fitted slightly better than the RSM model. In addition, in terms of fitting speed, the BP–ANN model achieved stability with a few cycles. Thus, the BP–ANN model had the dual advantages of faster fitting speed and higher precision. In the aspect of data prediction, the BP–ANN model had a better effect than the RSM model. When the number of BP–ANN data was sufficient, they can often present a better fitting and prediction effect than the RSM model. However, in this paper, the amount of data was 30 groups and 40 groups, respectively, indicating that the BP–ANN model can achieve better fitting and prediction results than the RSM model even with a small amount of data.

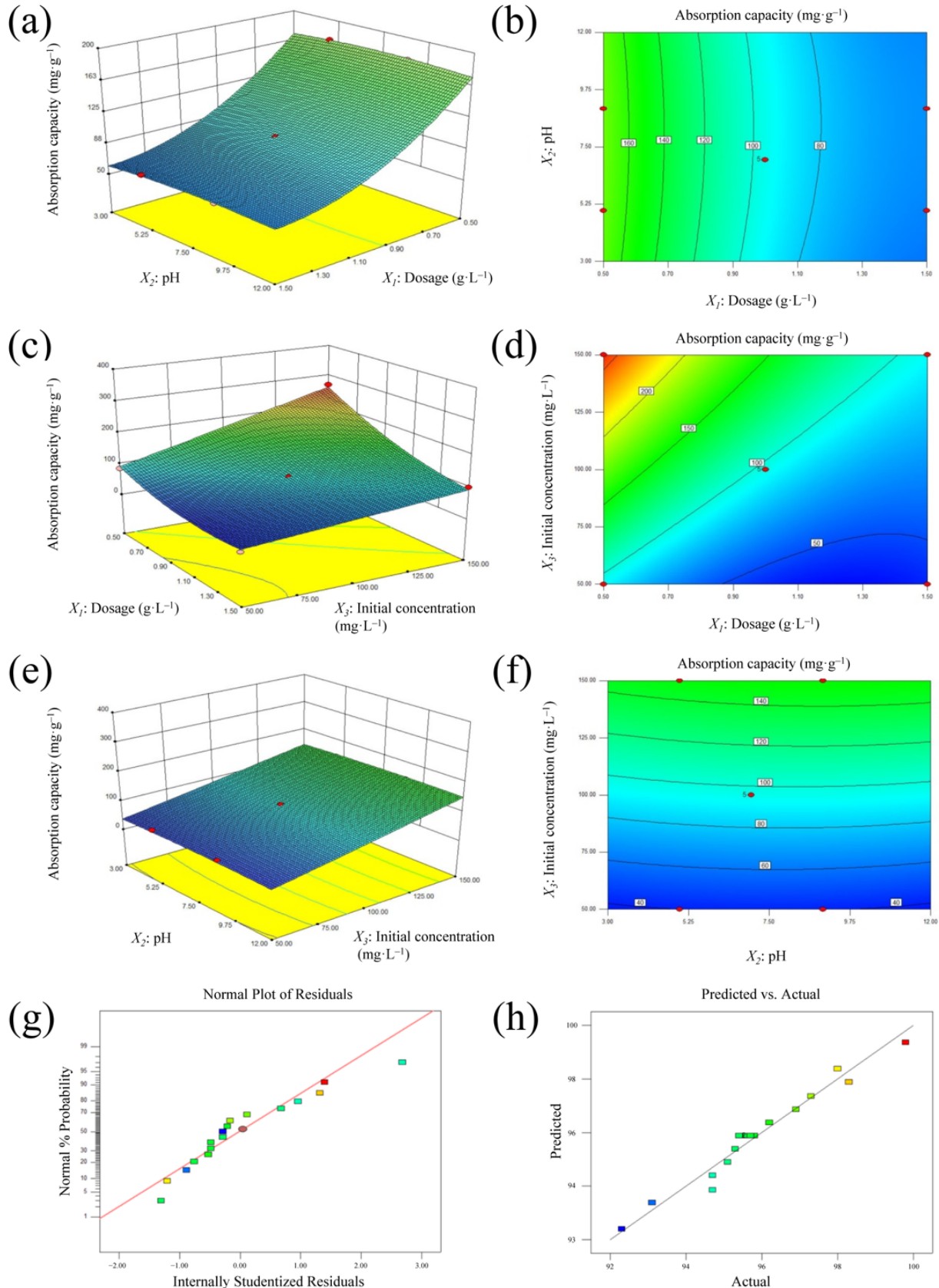

**Figure 4.** Duel effect on Cd(II) adsorption quantity, Normal Plot of Residuals, and Predicted vs. Actual values: response surfaces (**a**,**c**,**e**), contour plots (**b**,**d**,**f**), Normal Plot of Residuals (**g**), and Predicted vs. Actual values (**h**). Experimental conditions: [time] = 24 h, [temperature] = 25 °C.

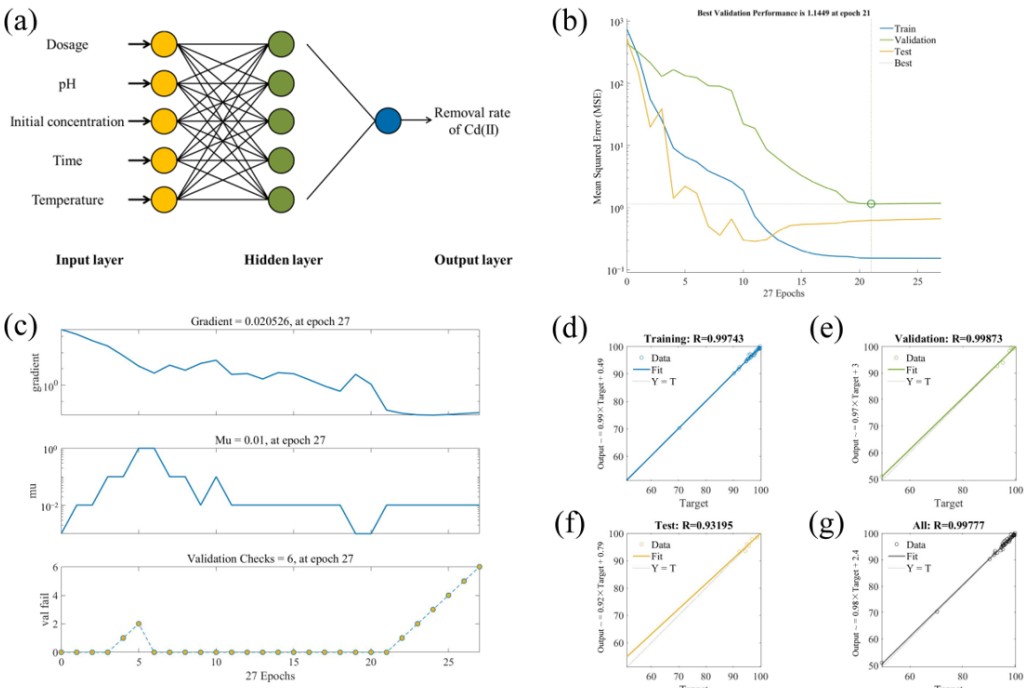

**Figure 5.** BP–ANN model of experimental data. Architecture of the BP–ANN model (**a**), error convergence curve during the iteration training of BP–ANN model (**b**), comparison between the output of BP–ANN model and the measured value (**c**), correlation between the experimental and simulated rate of removal during BP–ANN Training, Test, and Validation by MCB (**d**–**g**).

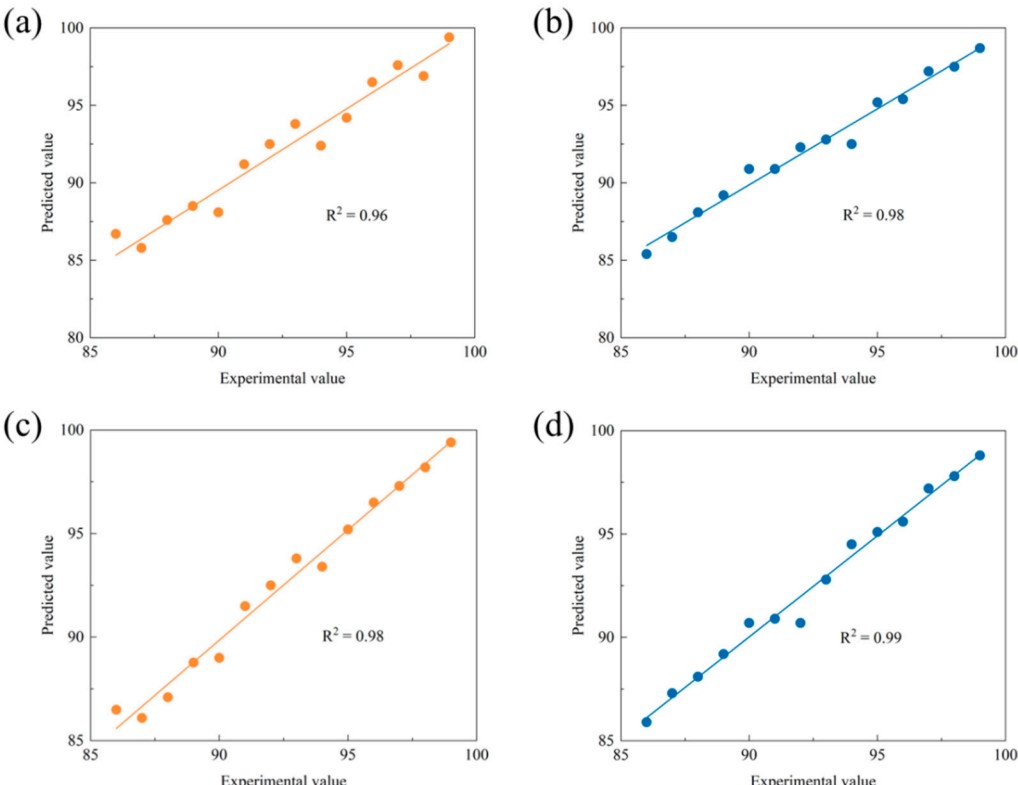

**Figure 6.** Comparison of the performances of the BP–ANN and RSM models. RSM (**a**) and BP–ANN (**b**) models for optimum preparation conditions, RSM (**c**) and BP–ANN (**d**) models for optimal reaction conditions.

*3.4. Characteristics*

3.4.1. Morphological Analysis

The basic elements of MCB were analyzed using an elemental analyzer. The carbon content was 51.41%, the oxygen content was 29.09%, the hydrogen content was 1.38%, and the nitrogen content was 0.55%. The H/C ratio was used to characterize the aromaticity of the adsorbent, and the O/C ratio was employed to gauge the level of carbonization [32]. Generally, the H/C ratio of CM biochar is 0.93 normally, while the MCB featured better aromatic and hydrophilic properties (0.93 vs. 0.03), which indicated that the MCB may have a better adsorption effect [33]. Moreover, the O/C ratio of MCB was 0.57, thus, the functional groups with oxygen were prevalent on the surface of MCB, which promoted adsorption.

The SSA of MCB was 37.43 $m^2 \cdot g^{-1}$ and the average pore size of MCB was 11.02 nm. The iron oxides loaded on the surface of MCB made the surface of the biochar rougher, so the SSA of MCB is larger than that of biochar without modification. A larger SSA may provide more opportunities for biochar to come into contact with Cd(II), resulting in enhanced physical adsorption capacity. The average pore size of MCB was small (11.02 nm), which may be because the iron oxides occupied part of the porous structure of MCB. Compared with the iron–modified rice straw biochar (8.28 $m^2 \cdot g^{-1}$, 25.83 nm) [34], the MCB prepared in this work had larger SSA and less pore size, which meant that MCB may have more excellent physical qualities. Moreover, the $N_2$ adsorption–desorption isotherms and pore size distribution curve were shown in Figure S1.

The SEM and SEM–EDS imagines of MCB were shown in Figure S2. There was porous tubular and carbon skeleton morphology in MCB via the scanning of the electron microscope, which may be formed from the straw fiber. Part of the carbon structure in MCB collapsed after 600 °C pyrolysis, while part of the carbon skeleton was still retained, which may be because the residence time of pyrolysis was only one hour. The surface of the MCB was slightly rough, and the retained carbon skeleton of MCB provided the basis for the physical adsorption including the pore filling of MCB.

The SEM–EDS images showed that the MCB was successfully loaded with Fe; thus, these images confirmed the conclusion in EA, that the content of Fe was high, achieving 78.51%. In addition, SEM–EDS images also showed that Fe existed in some microporous structures. Moreover, the MCB also contained abundant Ca and Mg, which were evenly distributed on the surface of the carbon skeleton of MCB.

3.4.2. XRD and FTIR Analysis

The XRD spectra of MCB was shown in Figure 7a. The analysis of XRD spectra was applied to determine the existence of crystalline minerals and analyze their structure of MCB. The fluctuating peaks of the typical amorphous carbon and graphite carbon structures were about 22° and 44°, respectively, and these peaks were common in biochar characterization, indicating that MCB belonged to biochar material [35]. But the amorphous carbon structure of MCB disappeared for the disappearance of these characteristic peaks, which indicated that the modification substantially changed the carbon structure in MCB [10]. This conjecture was mutually corroborated the SSA and SEM–EDS imagines mentioned above, and the modification affected the physical and chemical properties of MCB, then affected its adsorption of Cd(II). The $Fe_3O_4$ (PDF#89–2355) and $Fe_2O_3$ (PDF#87–1166) were loaded on MCB (Figure 7a), indicating the iron was loaded onto the MCB and the MCB was magnetic. The $Fe^0$ had strong reducibility and the peak at $2\theta = 45°$ indicated $Fe^0$ was loaded on MCB, which indicating the MCB was reductive slightly, and REDOX reaction may occur during the adsorption. Once the metal was loaded on MCB, the loaded metal would grow as the prominent active site, moreover, the doped heteroatom could work as active sites or induce other Lewis base sites on MCB to promote the catalysis process [36]; thus, modification may increase the number of active sites of MCB available and enhance the ability to adsorb Cd(II).

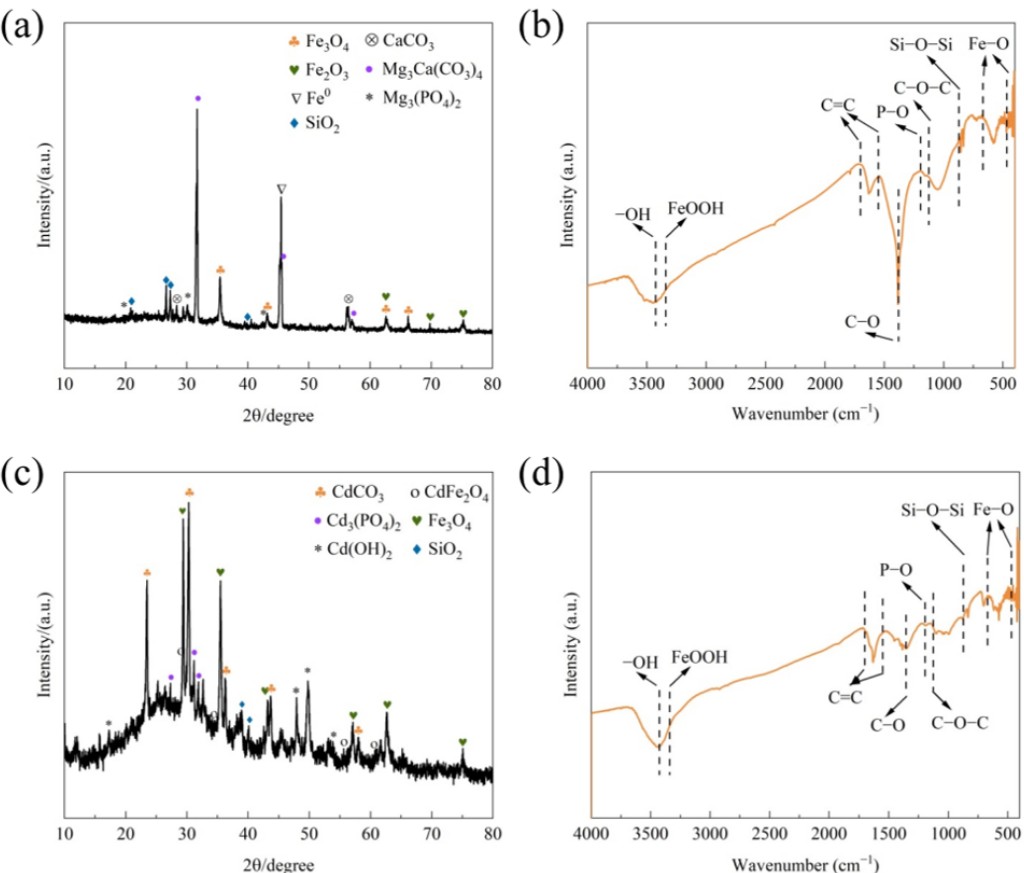

**Figure 7.** XRD and FTIR spectra of MCB (**a**,**b**) and Cd(II)–loaded MCB (**c**,**d**).

The MCB contained abundant Ca and Mg, which were evenly distributed on the surface of the carbon skeleton of MCB (Figure S2), and XRD spectra also confirmed the presence of metallic compounds on the surface of MCB (Figure 7a). The peaks indicated the presence of $CaCO_3$ (PDF#41–1475), $Mg_3(PO_4)_2$ (PDF#75–1491), and $Mg_3Ca(CO_3)_4$ (PDF#14–0409). These metal salts would react interfacially with Cd(II) in the solution on the surface of MCB, and the metal elements in the metal salts may be precipitated and undergo replacement reactions with the Cd(II) [37], resulting in substances with less solubility, such as $CdCO_3$ and $Cd_3(PO_4)_2$. Namely, Cd(II) may be removed via ion–exchange by MCB. Moreover, the MCB contained $SiO_2$ (PDF#45–1045), which corresponded to the results of EA.

FTIR spectra of MCB was shown in Figure 7b. The functional groups in MCB included –OH (3400 cm$^{-1}$), C=O (1735 cm$^{-1}$), Si–O–Si (873 cm$^{-1}$), and C–O–C (1099 cm$^{-1}$). The modification also affected the carbon structure of MCB. After the Fe modification, the peaks of –CH2 (2921 cm$^{-1}$), C–O, and –CH (946 and 873 cm$^{-1}$) in MCB were changed, thus, the modification may affect the carbon–containing functional groups. Though stretching vibrations of the aromatic C–C ring in MCB were retained after modification, the aromatic structure weakened due to Fe having invaded the carbon structure [38]. After modification, Fe–O (470–669 cm$^{-1}$) and FeOOH (3340 cm$^{-1}$) had been detected in MCB [39,40], which once again confirmed that iron was successfully loaded onto MCB. The vibration peaks at 927 and 786 cm$^{-1}$ were the aromatic and hetero–aromatic structure, respectively, which provided pi electrons. As the pi electrons had a potential advantage to capture heavy metal ions, MCB had the potential to adsorb Cd(II) via $\pi$ bonds. The vibrational transitions in the 1094–1182 cm$^{-1}$ region were due to P=O and P–O stretching vibrations of the P–O–P and P=O–OH groups, and those in the 900–926 cm$^{-1}$ region were ascribed to the asymmetric stretching of the P–O–P group [41], and this result was consistent with the results of XRD spectra analysis above, that MCB contained phosphate compounds.

### 3.4.3. Magnetic Properties

The hysteresis loop was used to characterize the response of magnetic materials, and the hysteresis loop of the MCB was displayed in Figure S3. The saturation magnetization value of the MCB was 12.8 emu·g$^{-1}$, which indicated that MCB can be rapidly magnetically separated from solution in a magnetic field environment.

### 3.5. Batch Experiments

### 3.5.1. Adsorption Kinetics and Isotherms

The fitting results were shown in Table S7 and Figure S4. The R$^2$ of the pseudo-first-order model (PFO), pseudo-second-order model (PSO), Elovich model, and internal diffusion model fitting for the adsorption of Cd(II) on MCB were 0.9848 (Figure S4a), 0.9999 (Figure S4a), 0.7268 (Figure S4c), and 0.9050 (Figure S4b). Thus, the PSO predicted the adsorptions of Cd(II) well, while the Elovich model could not fit the adsorption well. Moreover, the internal diffusion was not the sole rate-determining factor for the C = 8.6288 in the fitting results of the internal diffusion model. In short, the kinetic studies showed that the PSO could fit the experimental data well, which suggested that the Cd(II) adsorption by MCB was mainly based on chemisorption [42].

The R$^2$ of the Langmuir model and Freundlich model fitting for the adsorption of Cd(II) were 0.9956 and 0.9910 (Figure S4d). By comparing the R$^2$ values of the two models, the results showed that the Langmuir model better illustrated the interaction between MCB and Cd(II). The results implied that the absorption of MCB on Cd(II) was unimolecular [42], and the adsorption capacity was 612.43 mg·L$^{-1}$.

To sum up, the PSO and Langmuir models fit the adsorption process of Cd(II) on MCB better, and adsorption capacity was 612.43 mg·L$^{-1}$, indicating the Cd(II) adsorption by MCB was mainly based on chemisorption, and was unimolecular.

### 3.5.2. Adsorption Mechanisms

As shown in Figure S5, pH was also an important index affecting the adsorption effect of Cd(II) by MCB. With the pH of 3 and 5, the adsorption capacity was 62.4 and 85.4 mg·g$^{-1}$, respectively. When the reaction solution was acidic, the adsorption effect was poor, and the stronger the acid, the worse the adsorption effect. On the contrary, with the pH of 9 and 11, the adsorption capacity was 97.3 and 99.5 mg·g$^{-1}$, respectively. The form of Cd(II) in different pH reaction solutions may be varied. When pH $\leq$ 6, there were ions in a free state, like Cd$^{2+}$, and when pH was 6–9, there were ions existed in the form of Cd$^{2+}$ and Cd(OH)$_2$, while when pH $\geq$ 9, there were ions mainly existing in the form of Cd(OH)$_2$. When pH $\leq$ 3, there was electrostatic repulsion between MCB and Cd(II) for the surface of MBC was positively charged when the pH value of the Cd(II) solution was less than 3. Moreover, there was a lot of H$^+$ in the acidic solution, and the H$^+$ would compete with Cd(II) for adsorption sites on the surface of MCB, which reduced the adsorption effect.

The SEM and SEM–EDS imagines of Cd(II)-loaded MCB were shown in Figure S6, and the XRD spectra and FTIR spectra of Cd(II)-loaded MCB were shown in Figure 7. Compared with the MCB (Figures S2a and S6a), some pore structures of Cd(II)-loaded MCB were blocked with small solid agglomerates, which were also attached to the biochar skeleton. It was speculated that these solid agglomerates were the chemical precipitation of Cd(II). The EDS confirmed that the particles on biological carbon were Cd(II)-contained substances. These results indicated that MCB may adsorb Cd(II) via pore filling. As shown in Figure S6c,f, compared with the MCB, the presence of Cd(II) on the MCB surface suggests that the MCB had successfully loaded Cd(II) after absorption. According to the SEM–EDS imagines, Cd(II) was evenly distributed on the surface of Cd(II)-loaded MCB, and the distribution of Cd(II) was related to the distribution of Fe on the surface, indicating that the iron oxide on the surface of MCB may play a part in the adsorption on Cd(II).

As shown in Figure 7c, comparing the biochar before adsorption, the XRD spectra of Cd(II)-loaded MCB showed new peaks, such as CdCO$_3$ (PDF#42–1342), Cd$_3$(PO$_4$)$_2$ (PDF#72–1959), and Cd(OH)$_2$ (PDF#20–0179), determining that Cd(II) may be removed via

co–precipitation and ion-exchange. Moreover, iron oxides loaded on MCB also reacted with Cd(II) to form $CdFe_2O_4$ (PDF#22–1063). Figure 7d showed the FTIR spectra of Cd(II)-loaded MCB. The broad peak band from 630 to 750 $cm^{-1}$ was mainly metal oxygen-containing functional groups, such as Fe–O, while after Cd(II) adsorption, these broad peak bands mentioned above decreased, indicating that Cd(II) reacted with the oxygenic groups. After Cd(II) adsorption, the intensity of peaks at 1573, 1398, 1029, 981, and 800–700 $cm^{-1}$ changed slightly, which indicated that C=C, C=O, C–O, and other groups participated in the adsorption reaction, which determining qualitatively that cation–π interaction existed in the adsorption [43].

The XPS spectra were displayed in Figure S7. The C 1s fine spectrum of the sample consisted of three peaks, C–C (284 eV), C–O–C (286 eV), and O–C=O/C=O (289 eV). While after adsorption, the peaks of C 1 shifted, which indicated the structure changed after adsorption, and oxygen-contained functional groups made a contribution to the Cd(II) adsorption. The peak positions for Cd 3d were 406 and 414 eV, whose percentages were 47.6 and 52.4%, respectively. There was no REDOX reaction involved for the valence of Cd(II) did not change. Combined with the XRD analysis mentioned above, Fe(II) and Fe(III) compounds were loaded on the surface and their contents were 73.4 and 26.6%, respectively. As shown in Figure S7b, the peak positions for Fe 2p were 710, 711, 717, 723, and 725 eV. The Fe 2p peaks were changed after absorption, showing that presence of Cd(II) caused the variation in Fe 2p. Besides, compared with Fe 2p before adsorption (Figure S8), the contents of $Fe^0$ and $Fe^{2+}$ loaded on Cd(II)-loaded MCB decreased, thus, $Fe^0$ and $Fe^{2+}$ may were oxidized during the absorption. The Fe(III) loaded on MCB interacted with the anions (such as carbonate and phosphate) exited from MCB. The hydrolysis of Fe(III) made the solution acidic, and $H^+$ replaced Mg(II) in MCB, resulting in that a Si–O–Mg–O–Si group broke down into two Si–OH groups, so a large number of Si–OH groups were generated on the surface of MCB [43]. Moreover, the Si–OH group can have the following surface coordination adsorption with Cd(II) [44]. The reaction process was shown in Equation (6) to Equation (8).

$$SiOH + Cd^{2+} \rightarrow SiOCd^+ + H^+ \tag{6}$$

$$2SiOH + Cd^{2+} \rightarrow (SiO)_2Cd + 2H^+ \tag{7}$$

$$SiOH + Cd^{2+} + H_2O \rightarrow SiOCdOH + 2H^+ \tag{8}$$

After adsorption, the characteristic peak of $Fe_3O_4$ (30.1°) was still observed, while the characteristic peak of $Fe^0$ disappeared, indicating that Fe loaded on MCB was oxidized [45,46], which was shown as Equation (9) to Equation (12).

$$Fe^0 \rightarrow Fe(OH)_2 \rightarrow Fe_3O_4 \rightarrow (\gamma\text{-}Fe_2O_3) \tag{9}$$

$$2Fe^0 + 4H^+ + O_2 \rightarrow 2Fe^{2+} + 2H_2O \tag{10}$$

$$Fe^{2+} + H_2O \rightarrow Fe^{3+} + 1/2H_2 + OH^- \tag{11}$$

$$Fe^{3+} + 3OH^- \rightarrow Fe(OH)_3 \tag{12}$$

To sum up, the adsorption was dominated via chemisorption with the mechanisms of ion-exchange, electrostatic attraction, pore-filling, co-precipitation, and formation of complexations (Figure 8).

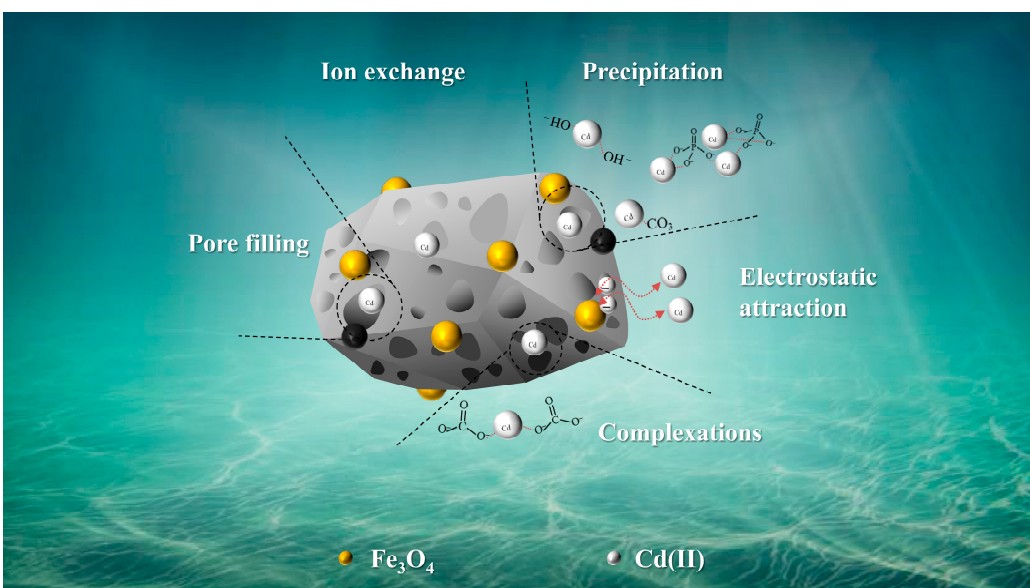

**Figure 8.** Surface interaction between Cd(II) and MCB functional materials.

## 4. Conclusions

In this work, MCB was prepared via cow manure, and employed for Cd(II) removal. PT, HT, and iron content are the key factors of MCB preparation, while the optimal conditions were obtained by the RSM method as 160 °C for HT, 600 °C for pyrolysis, and Fe loading with 10 wt%, and experimental parameters were optimized as 1 g·L$^{-1}$ for dosage, 7 for pH, and 100 mg·L$^{-1}$ for initial concentration. The BP–ANN model fit better than the RSM model, with lower error values. The constructed BP–ANN model overcame the shortage of large amounts of data training to achieve high precision, demonstrating better contingency and stability. As for thermodynamics, the PSO and the Langmuir model could better mimic the surface interactions between Cd(II) and MCB. The adsorption was dominated by chemisorption with the mechanisms of ion-exchange, electrostatic attraction, pore-filling, co-precipitation, and formation of complexes. All these results demonstrated the potential of cow manure as feasible biomass feedstock for functional materials preparation and wastewater purification.

**Supplementary Materials:** The following supporting information can be downloaded at: https://www.mdpi.com/article/10.3390/pr11082295/s1, Text S1: Calculation of adsorption capacity; Table S1: Variables and experimental conditions for RSM; Table S2: BBD experimental design with the actual values of adsorption quantity with different biochar; Table S3: ANOVA for response surface quadratic model; Table S4: BBD experimental design with the actual values of adsorption quantity under different reaction conditions; Table S5: ANOVA for response surface quadratic model; Table S6: Comparison of the statistical parameters of ANN and RSM models for the various responses; Table S7: Fitting parameters of kinetic model and isotherm model of Cd(II) absorption; Figure S1: N$_2$ adsorption–desorption isotherms and pore size distribution curve; Figure S2: SEM (a and b) and SEM–EDS imagines of MCB (c – f); Figure S3: Magnetic hysteresis loop of MCB; Figure S4: Fitting–figures of PFO, PSO (a), internal diffusion model (b), Elovich model (c), and isotherms (d) for the adsorption of Cd(II) on MCB. Experimental conditions: [initial concentration of Cd(II)] = 100 mg· L$^{-1}$, [dosage] = 1 g·L$^{-1}$, [time] = 24 h, [temperature] = 25 °C; Figure S5: The effect of pH for adsorption Cd(II) by MCB. Experimental conditions: [initial concentration of Cd(II)] = 100 mg· L$^{-1}$, [dosage] = 1 g·L$^{-1}$, [time] = 24 h, [temperature] = 25 °C; Figure S6: SEM–EDS imagines of Cd(II)–loaded MCB; Figure S7: XPS spectrums of Cd(II)–loaded MCB; Figure S8: XPS Fe *2p* spectrum of MCB.

**Author Contributions:** Y.W.: formal analysis, original draft; D.C.: data curation, software, methodology; Y.Z.: data curation, investigation, methodology; H.W.: supervision, writing—review and editing; R.X.: funding acquisition, resources, supervision. All authors have read and agreed to the published version of the manuscript.

**Funding:** This work was supported by the National Natural Science Foundation of China [22264025, 52100147], Applied Basic Research Foundation of Yunnan Province [202201AS070020, 202201AU070061], and the Science and Technology Research Project of Education Department of Jiangxi Province [DA202102159].

**Data Availability Statement:** Not applicable.

**Conflicts of Interest:** The authors declare no conflict of interest.

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
