# Peer review of "Cadmium Elimination via Magnetic Biochar Derived from Cow Manure: Parameter Optimization and Mechanism Insights"

_processes, doi:10.3390/pr11082295_

Round 1
Reviewer 1 Report
please check the attached file

An extensive English grammar and spell check is needed for the whole manuscript
Reviewer 2 Report
Review report
The authors have stated that they have produced Magnetic biochar using cow manure nd iron salt by hydrothermal method synthesis techniques. They have employed synthesized materials in the removal of Cd ion via adsorption method. Moreover, authors have applied RSM method along with back propagation-artificial neural network (BP-ANN). The procedure and parameters were optimized using Box-Behnken model. However, there are lot of issues with the manuscript. I recommend major revision. Followings are my comments and suggestions.
1. There are lot of typo and grammatical mistakes in abstract. Abstract must be rewritten and must be focused on results findings.
2. I advised that manuscript language must be improved with the help of a native speaker, as there are lot of grammatical errors are in introduction section as well.
3. In the synthesis section, as the iron salts Fe2+ and Fe3+ are used and mixed together along with NaOH, it must be formation of Fe3O4 or Fe2O3 nanoparticles. Authors must clear this.
4. Authors have stated that the prepared adsorbent was analyzed by various techniques. Authors claimed that Fe metals are dominated in CM but as XRD FTIR XPS and other data shows the formation of Fe3O4 as well.
5. Authors must clarify the form of iron whether it is metal or oxide of Fe, I advised it must be doing RAMAN technique as well.
6. The figures quality is very poor; it must be improved.
7. Where is the analysis of removal explained and how it was analyzed experimentally?
Extensive editing of English is required. It must be improved with the help of a native speaker.
Reviewer 3 Report
The manuscript (processes-2493370) entitled " Cadmium elimination by magnetic biochar derived from cow manure: Parameters optimization and mechanisms insights" by Wang and Xu et al. reported the synthesis and application of magnetic biochar derived from cow manure.
1. What is the main question addressed by the research?
The synthesis and application of magnetic biochar derived from cow manure.
2. Do you consider the topic original or relevant in the field? Yes.
Does it address a specific gap in the field? To some extent.
3. What does it add to the subject area compared with other published material?
The work demonstrated the potential application of CM for Cd(II) removal and optimized by machine learning processes.
4. What specific improvements should the authors consider regarding the methodology? What further controls should be considered?
The characterization of the prepared materials and controls are missing. Such as specific surface area measurement, SEM.
5. Are the conclusions consistent with the evidence and arguments presented and do they address the main question posed?
Not good enough. Besides the machine learning processes and model, the typical structure-property correlation should be explored and discussed.
6. Are the references appropriate? Yes
7. Please include any additional comments on the tables and figures.
The manuscript needs minor language proofreading, for example, in 2.2. Synthesis of magnetic CM biochar, “The three mixtures were placed in hydrothermal synthesis reactors severally and heated at different HT……”
Moderate editing of English language required
Round 2
Reviewer 2 Report
The authors have done all the comments. I recommend for publication.
Author Response
Thank you very much.
Reviewer 3 Report
After revision, I think the manuscript is in good shape for publication.
Minor editing of English language required.
